# Intense Pulsed Light Attenuates UV-Induced Hyperimmune Response and Pigmentation in Human Skin Cells

**DOI:** 10.3390/ijms22063173

**Published:** 2021-03-20

**Authors:** Juewon Kim, Jeongin Lee, Hyunjung Choi

**Affiliations:** R&D Unit, Amorepacific Corporation, Yongin 17074, Korea; mizzle23@amorepacific.com (J.L.); heart@amorepacific.com (H.C.)

**Keywords:** intense pulsed light, inflammation, melanogenesis, oxidative stress, antioxidative enzyme activity, photoaging

## Abstract

The skin of an organism is affected by various environmental factors and fights against aging stress via mechanical and biochemical responses. Photoaging induced by ultraviolet B (UVB) irradiation is common and is the most vital factor in the senescence phenotype of skin, and so, suppression of UVB stress-induced damage is critical. To lessen the UVB-induced hyperimmune response and hyperpigmentation, we investigated the ameliorative effects of intense pulsed light (IPL) treatment on the photoaged phenotype of skin cells. Normal human epidermal keratinocytes and human epidermal melanocytes were exposed to 20 mJ/cm^2^ of UVB. After UVB irradiation, the cells were treated with green (525–530 nm) and yellow (585–592 nm) IPL at various time points prior to the harvest step. Subsequently, various signs of excessive immune response, including expression of proinflammatory and melanogenic genes and proteins, cellular oxidative stress level, and antioxidative enzyme activity, were examined. We found that IPL treatment reduced excessive cutaneous immune reactions by suppressing UVB-induced proinflammatory cytokine expression. IPL treatment prevented hyperpigmentation, and combined treatment with green and yellow IPL synergistically attenuated both processes. IPL treatment may exert protective effects against UVB injury in skin cells by attenuating inflammatory cytokine and melanogenic gene overexpression, possibly by reducing intracellular oxidative stress. IPL treatment also preserves antioxidative enzyme activity under UVB irradiation. This study suggests that IPL treatment is a useful strategy against photoaging, and provides evidence supporting clinical approaches with non-invasive light therapy.

## 1. Introduction

The skin functions as the outermost organ and a defense barrier in the body. Skin aging is pivotal for extra stimulation defense and also protects against visible signs of senescence. Aging stress in the skin is provoked mainly by ultraviolet B (UVB) irradiation. UVB irradiation causes excessive reactive oxygen species (ROS) accumulation and extracellular matrix (ECM) degradation, and these factors result in accumulation of oxidative stress and nonfunctional elastin and collagen fibrils [1]. UVB-induced photoaging is an explicit example of aging stress, as indicated by comparisons between photoexposed (face and arm) and photoprotected (buttocks) skin; such comparisons have revealed that genes related to deoxyribonucleic acid (DNA) repair and replication, chromatin remodeling, oxidative stress responses, autophagy, protein metabolism, and cell growth and survival are differentially expressed in youthful skin compared with photoaged skin [2]. UVB light is absorbed by DNA and induces direct DNA damage, causes oxidative stress, and upregulates transcription factors [3]. UVB exposure elevates the generation of a variety of inflammatory cytokines and diverse genes associated with DNA repair, oxidative stress responses, protein metabolism, and immune responses. In some instances, accumulated damage caused by acute or spontaneous UVB exposure leads to a hyperimmune reaction in skin cells, and this hyperinflammatory immune response impairs health via hyperinflammatory immunodeficiency, which leads to further multiorgan damage [4].

UVB exposure is also associated with a risk of excessive melanin formation, which can lead to melasma. Melanin is found in the epidermis and protects DNA and skin cell organelles from UV irradiation. Melanin is generated upon UV exposure and forms a layer of pigmentation that protects the skin from UVA and UVB irradiation. However, acute UVB irradiation stimulates the epidermal melanogenic system, resulting in dysfunction of melanin formation and accumulation. This manifests clinically as inconsistent pigmentation, with regions of hyperpigmentation. Aberrant melanin production due to photoaging can be harmful to skin health [5].

Non-invasive treatment for amelioration of the skin aging phenotype has been studied and is becoming a useful therapeutic option because of its safety and because it does not require direct manipulation. Light-based technologies enable clinical application of light as a treatment for particular diseases and have improved the ability to effectively and safely attenuate skin aging [6]. Since laser and light therapy was first examined for use in the treatment of cutaneous disorders, many clinical trials have been conducted with different lasers and therapeutic strategies and an immense range of efficacies and harmful effects have been reported for light-based treatment of various skin disorders [7]. Although UV-based light treatment is used for inflammatory skin disorders such as psoriasis and atopic dermatitis, visible-light phototherapy is more accepted because it achieves positive outcomes such as immunomodulation, wound healing, inflammation mitigation, and tissue regeneration [8,9,10,11]. Intense pulsed light (IPL) therapy uses a light source that emits noncoherent light with a wavelength of 515–1200 nm. Similar to laser treatment, the basic purpose of IPL is to cause selective thermal damage to a target, and most comparative studies have certified that IPL and lasers show similar effectiveness [12]. In addition to being validated, IPL treatment has been found to be superior to laser therapy because it uses wavelengths that can penetrate diverse levels of the skin and target both the epidermis and the dermis simultaneously. The head of IPL is also larger than most laser spots, and IPL has a millisecond-range pulse duration, which allows for agile treatment of large areas and reduces the risk of heat-related inflammation due to enhanced thermal diffusion.

Given these advantages of IPL treatment, we investigated whether IPL therapy can attenuate UVB-induced aging stress related to hyperimmune reactions and hyperpigmentation in human skin cells. Yellow light attenuates excessive inflammatory reactions in vitro and in vivo, including in humans [13,14,15]. It has also been reported that green light ameliorates pigmented lesions and reduces postlaser pigmentation [16,17,18]. To investigate these light irradiation effects, we generated an IPL device with green (525–530 nm) and yellow (585–592 nm) light sources of 150 lumens with a 100 ms on/off pulse duration for skin cell treatment. We examined the ameliorative effects of IPL irradiation on UV-induced hyperinflammation and pigmentation in human epidermal keratinocyte and melanoma cell lines, in addition to investigating the molecular mechanisms related to inflammatory or melanogenic genes. According to our findings, we suggest that green and yellow IPL treatment is useful for UVB-induced skin damage or disorders, with the potential to reduce hyperinflammation and hyperpigmentation.

## 2. Results

### 2.1. IPL Device Outputs

The wavelengths of the light-emitting diode (LED) in the IPL device are 585–592.5 nm (yellow) and 525–530 nm (green). The device also features a handheld display unit and a sensor unit (Figure 1A). To measure the outputs of IPL, a small integrating sphere system and a high-accuracy spectroradiometer (HAAS-3000, EVERLIGHT, Taiwan) were used (Figure 1B). Analyses revealed that the total voltage and current applied for yellow IPL were 10.5 V and 200 mA, respectively, while those for green IPL were 14.3 V and 100 mA, respectively. The device applies a 100 ms on/off pulse and emits 20 pulses of light in that time. In addition, the voltage, current, pulse duration, and number of pulses can be adjusted. A cell test was performed in which 230 lumens, 448 mW, and 0.9 J of energy (spot size 6.2 cm^2^) were repeatedly applied at 30 min intervals. An average deviation of 28% from the manufacturer’s IPL output settings was observed, with the greatest conflict occurring at the lowest IPL setting (Figure 1C, Appendix A). Moreover, the maximal parity between the IPL settings and the measured outputs occurred at the highest IPL setting. Temperature changes throughout the cell treatment system during irradiation with IPL were also evaluated continuously (Appendix A). Because the device applies a 100 ms on/off pulse duration, the temperature change degree due to sequential irradiation did not seriously affect IPL-treated cells (Appendix A). In addition, there was no significant change in temperature when IPL was used to irradiate swine skin, which is similar to human skin, for up to 5.21 h (Appendix A). The process of IPL study is shown in Figure 1D.

### 2.2. IPL Treatment Attenuates the UV-Induced Hyperimmune Reaction

Under exposure to UVB irradiation in the epidermis, skin cells, especially keratinocytes, produce proinflammatory cytokines, which leads to skin inflammation [19,20,21,22]. To evaluate the effect of UVB radiation on the immune response, the expression levels of cytokines were measured after 18 h of UVB irradiation (20 mJ/cm^2^). We set the UVB irradiation conditions based on data on cell viability and the levels of a representative UV irradiation-induced inflammatory cytokine interleukin (IL)-6 [23,24] (Appendix A). In a human cytokine assay, exposure of normal human epidermal keratinocytes (NHEKs) to UVB irradiation significantly induced the generation of inflammatory cytokines, including IL-6, IL-8, and tumor necrosis factor receptor 1 (TNFR1), and reduced the levels of the adipokine adiponectin, which is related to wound healing and skin anti-inflammation (Figure 2A–C) [25,26]. Exposure to UVB also increased the expression of IL-7, transforming growth factor beta (TGF-β), and interferon gamma (IFN-γ) (Figure 2A–C). These data show that UVB irradiation induces inflammatory reactions also in gene expression levels of keratinocytes (Figure 2D). To examine the effects of IPL on these hyperimmune reactions, we treated UVB-irradiated NHEKs with yellow, green, or combined yellow/green IPL at 150 lumens with a 100 ms pulse duration and a 30 min interval time for 6 to 24 h according to previously reported IPL cell culture treatment protocols [27]. Treatment with either yellow or green IPL did not affect cell viability under these conditions (Appendix A), but yellow IPL attenuated the UVB-induced increases in IL-6 expression levels after 18 and 24 h of treatment (Appendix A). According to these results, we performed IPL treatment with settings of 150 lumens, a 100 ms pulse duration, and a 30 min interval time for 18 h to estimate the effects of IPL against UVB-induced active immune reactions and to confirm that IPL treatment does not elicit cytotoxicity. As shown in the results, yellow IPL treatment significantly reduced the elevations in inflammatory cytokine expression and recovered adiponectin expression (Figure 2A–D). Although green IPL did not exert notable effects, combined treatment with yellow and green IPL was obviously more effective than yellow IPL irradiation alone in reducing UVB-induced inflammatory gene expression (Figure 2). These results were also confirmed at the protein level with supernatants of UVB- and IPL-treated samples. We determined that combined treatment with yellow and green IPL attenuated the increases in IL-6, IL-7, and IL-8 secretion by NHEKs after UVB stimulation (Figure 3A–C). Moreover, we determined that IPL irradiation decreases the protein expression of TNFR1, TGF-β, the TGF-β receptor, IFN-γ, and IL-12 and enhances that of adiponectin (Figure 3D).

### 2.3. IPL Treatment Reduces UV-Induced Hyperpigmentation

IPL has been used therapeutically in a clinical setting to reduce skin pigmentation in pigmented lesions and to decrease postlaser pigmentation [16,18,28]. The settings in these cases were similar to the yellow or green IPL settings, and the IPL treatment was conducted under conditions of elevated melanogenesis. Since the effects of IPL on skin pigmentation have been studied only in the clinic and since the mechanism of action has not been uncovered, we investigated the effects of yellow and green IPL on post-inflammatory hyperpigmentation in human epidermal melanocytes at the cellular level. UVB irradiation greatly increased melanin production by enhancing the expression of representative melanogenic genes, including microphthalmia-associated transcription factor (MITF), tyrosinase (TYR), tyrosinase-related protein 1 (TYRP1), and dopachrome tautomerase (DCT) (Figure 4A,B). We confirmed that yellow or green IPL treatment reduces UVB-induced melanin accumulation and that combined treatment with yellow and green IPL decreases melanin production (Figure 4A). Melanin generation in the yellow, green, and combined yellow + green groups was 60.7%, 38.4%, and 75.4% lower than that in the UVB-irradiated group, respectively. Interestingly, the changes in melanogenic gene expression did not clearly correlate with the melanin levels under IPL treatment (Figure 4B). Green IPL decreased the gene expression of MITF, TYR, TYRP1, DCT, and melanoma-associated antigen recognized by T cells 1 (MART1) more than yellow IPL, even though melanin levels were lower in the yellow IPL group than in the green IPL group (Figure 4B). This discrepancy may have been due to specific regulation of tyrosine protein kinase KIT (cKIT), SRY-box transcription factor 10 (SOX10), and stem cell factor (SCF) by yellow IPL irradiation (Figure 4B). Yellow IPL clearly attenuated tyrosinase activity more strongly than green IPL, and combined application of yellow and green IPL exerted additive effects on UVB-induced hyperpigmentation (Figure 5A). In addition, we found that IPL treatment reduced the expression of the melanogenic proteins MITF, TYR, TYRP1, DCT, SCF, and SOX10, which are involved in UV-induced hyperpigmentation (Figure 5B).

### 2.4. IPL Treatment Attenuates ROS Accumulation and Preserves Cellular Antioxidative Enzyme Capacity after UVB Irradiation

To investigate the associated cellular events that may be involved in the modulation of immunity and melanogenesis by IPL, the intracellular levels of ROS were examined using 2’,7’-dichlorofluorescein diacetate (DCFDA), a redox-sensitive dye. Combined treatment with yellow and green IPL attenuated the UVB-induced ROS elevations, returning the ROS levels to control levels (Figure 6A). IPL treatment also increased resistance to oxidative stress under UVB-treated conditions (Figure 6B). Interestingly, IPL did not exert a ROS-reducing effect or enhance oxidative stress resistance under normal conditions (Figure 6A,B). On the other hand, IPL preserved the activity of the cellular antioxidative enzymes superoxide dismutase (SOD) and catalase, which was significantly reduced after UVB irradiation (Figure 6C,D). UVB irradiation decreased the SOD capacity to 32.8 ± 9.3% of the control level, while yellow and green IPL recovered it to 72.5 ± 5.3% of the control level (Figure 6C). Catalase activity similarly declined to 27.5 ± 1.9% of the control level after UVB treatment, but combined IPL treatment restored it to 66.8 ± 2.5% of the control level (Figure 6D). These results demonstrate that IPL treatment attenuates ROS accumulation by restoring antioxidant capacity and elevating resistance to oxidative stress, although it does not exert notable effects under non-irradiated conditions.

## 3. Discussion

UV irradiation, especially UVB irradiation (280–320 nm), induces cutaneous inflammation and pigmentation [29,30]. These changes are caused by DNA mutations and alterations in the expression of immune response- and melanogenesis-related genes. In this study, cytokine and gene expression analyses revealed that UVB stimulation increases inflammatory and melanogenic gene expression levels in human keratinocytes (Figure 2 and Figure 4). To prevent or reduce such negative effects of UVB irradiation, many non-invasive treatments using light have been attempted [7,9,31]. However, the mechanisms of action of these light-based treatments are largely known. In the present study, we evaluated the effects of IPL on UVB-induced hyperimmune and melanogenic reactions in NHEKs and investigated the mechanism underlying the effects of IPL treatment on proinflammatory cytokines and melanogenic genes. Our results suggested that UVB radiation upregulates proinflammatory cytokine expression and that yellow IPL attenuates this UVB-induced abnormal reaction (Figure 2). IPL-mediated inhibition of the UVB-induced increases in cytokine levels reduced IL-6, IL-7, and IL-8 production (Figure 3). UVB-induced activation of proinflammatory cytokines was alleviated by inhibition of interleukin expression via yellow IPL application. Although green IPL did not demonstrate considerable efficacy against the UVB-induced immune response, combined treatment with yellow and green IPL resulted in additional suppression of hyperimmune responses (Figure 2 and Figure 3). This effect may have been due to the fact that yellow IPL has direct inhibitory effects on skin cell inflammation that differ from those of green IPL [13,14] and to the possibility that IPL may improve the responsiveness of several photoreceptors that react to green light [32,33,34,35,36]. Our findings provide remarkable insight into capable therapies for UVB-induced skin hyperimmunity.

The potential usefulness of light-based therapy has been studied for the treatment of photoinduced pigmentation and melasma accompanying various conditions [16,27,34,37,38]. In the current study, IPL treatment reduced UVB-induced melanin accumulation; treatment with yellow IPL restored accumulation to the control level, and combined treatment with green and yellow IPL had dramatic additive effects (Figure 4A). Interestingly, green IPL decreased the gene expression of MITF, TYR, TYRP1, DCT, and MART1 more than yellow IPL, despite resulting in lower melanogenesis inhibition (Figure 4B). This outcome may have occurred because yellow IPL specifically regulated cKIT, SOX10, and SCF, which, in turn, induced production of cellular signaling molecules and acted synergistically with other cytokines, probably interleukins. Moreover, the activity of the main enzyme in melanogenesis, tyrosinase, was greatly downregulated after yellow IPL treatment (Figure 5A). Melanin production is an oxidizing process; therefore, the cellular redox state affects melanin accumulation. The elevations in ROS levels induced by UVB irradiation were largely reduced by yellow IPL and combined yellow and green IPL (Figure 6A). These results demonstrate that IPL could be a useful tool for alleviation of the hypermelanogenesis induced by UVB radiation.

Previous studies have suggested that ROS intermediates signaling events leading to gene expression after UVB radiation [1,3,4,5,19,39]. Excessive exposure to UVB radiation leads to excessive oxidative stress and results in damage and disorders in skin cells [1,3,4,39]. UVB-induced hyperimmune responses in keratinocytes and melanogenic responses in melanocytes are related to cellular redox levels [40,41,42], and these changes affect each other; thus, we investigated whether oxidative stress resistance and antioxidant capacity are altered by IPL manipulation. As shown in Figure 6B, IPL application not only reduced cellular ROS levels but also increased resistance against oxidative stress. Furthermore, IPL treatment preserved the reduction in innate antioxidant activity induced by UVB radiation (Figure 6C,D). We propose that IPL, if used appropriately, can effectively restore unusual redox states to normal in skin cells and regulate the immune response and melanogenesis.

IPL has been used therapeutically in various clinical settings and has been shown to have photobiomodulatory effects on connective tissue cells. The IPL settings applied in this study involved polychromatic light at wavelengths ranging from 525 to 592.5 nm. Light at such wavelengths is able to penetrate various chromophores at different depths in biological cells and tissues [12,43]. Although IPL application outputs have lacked consistency because of the diversity of targets and light source parameters, in vitro cellular models are pivotal tools for preclinical evaluation of treatment modalities and for parameter optimization [27,44,45]. In addition, in vitro models are important tools for enhancing the understanding of molecular mechanisms, eventually enabling the development of improved applications. Indeed, using a skin cell model and an IPL device, we found that yellow, green, and combined yellow and green IPL can potently attenuate UVB-induced excessive immune responses and melanin accumulation. Moreover, our findings suggest that the mechanisms of action of IPL treatment involve skin cell inflammation and pigmentation. Taken together, our findings validate manipulation of IPL as a useful strategy for treatment of UVB-induced hyperimmune reactions and melanogenesis in the skin.

## 4. Materials and Methods

### 4.1. Chemicals and Reagents

Antibodies against MITF (C5), tyrosinase (H-109), TYRP1 (H-90), DCT (C-9), IL-12 (JJ-07), and GAPDH (FL-335) were obtained from Santa Cruz Biotechnology (Santa Cruz, CA, USA). TNFR1 (C25C1), TGF-β (#3711), TGF-β RII (#79424), IFN-γ (3F1E3), SCF (#2093), SOX10 (#89356), and adiponectin (#2789) antibodies were purchased from Cell Signaling Technology (Cambridge, MA, USA). Secondary antibodies for Western blotting were obtained from Cell Signaling Technology.

### 4.2. Cell Culture and Growth Conditions

NHEKs obtained from neonatal foreskin were purchased from Lonza (Basel, Switzerland) and cultured in keratinocyte growth medium (KBM Gold) with BulletKit (Lonza, Walkersville, MD, USA) containing insulin, human epidermal growth factor, bovine pituitary extract, hydrocortisone, epinephrine, transferrin, and gentamicin/amphotericin B. The cells were serially passaged until 70–80% confluence was achieved, which was no more than three times. Cell proliferation and cytotoxicity were measured with a Cell Counting Kit-8 (Dojindo, Kumamoto, Japan). Moderately pigmented human epidermal melanocytes were purchased from Cascade Biologics (Portland, OR, USA) and maintained and passaged in Medium 254 (#M254500) supplemented with Human Melanocyte Growth Supplement (Life Technologies, Carlsbad, CA, USA), 10% fetal bovine serum, 100 U/mL of penicillin G, and 100 μg/mL of streptomycin sulfate. The human epidermal melanocytes were incubated at 37 °C with 5% CO_2_ and regularly passaged at a density of 80% (1:8 ratio). Cell viability was measured using the Cell Counting Kit-8 (Dojindo, Kumamoto, Japan).

### 4.3. UVB Light Apparatus

We used a Biosun UV irradiation system with a lamp that produces wavelengths at approximately 280–320 nm (*λ*_max_: 312 nm) (Vilber Lourmat, Marnes-la-Vallee, France) to generate UVB radiation. The apparatus had a culture dish tray, and the temperature did not exceed 30 °C during exposure. Before UVB irradiation, the cells were washed with 1 mL of phosphate-buffered saline (PBS), and 0.5 mL of fresh PBS was added to each well. The cells were irradiated at 20 mJ/cm^2^ without the plastic dish lid. After UVB irradiation, the cells were incubated again in basal medium and treatments were performed at various time points prior to the harvest step.

### 4.4. IPL Treatment

NHEKs and human epidermal melanocytes were treated with IPL (Everlight Electronics, New Taipei city, Taiwan) using 525–530 nm and 585–592.5 nm cutoff filters with a pulse duration of 100 ms and a flux of 220 to 250 lumens. The device used 10 yellow and green LEDs each and produced 20 emission bursts at a time. Cells in the growth phase were obtained and cultured at 37 °C under 5% CO_2_ and saturated humidity. When cell adherence and fusion reached 80% in the culture flask (6 cm^2^), the supernatant was discarded and the adherent cells were washed twice with PBS. The cells were then irradiated using IPL after UVB irradiation. Each experimental group had four replicate wells. Detection was carried out 18 h post-irradiation with a 30 min irradiation interval; the detection time points had been determined in previous preliminary experiments. To measure the outputs of the IPL source, EVERFINE’s small integrating sphere system and a HAAS-3000 spectroradiometer were used (Everlight Electronics, New Taipei city, Taiwan). Cells were obtained at corresponding time points, the supernatant was discarded, and the cells were washed twice with PBS.

### 4.5. Cytokine Array and ELISA

The medium was harvested from NHEKs, stimulated with UVB radiation, and centrifuged for 15 min at 4 °C. The supernatants were freeze-dried and used for multiple cytokine measurements with Human Cytokine Array C3 and Human Cytokine Array C4 Kits (RayBiotech, Norcross, GA, USA) according to the manufacturer’s instructions. The arrays were imaged with a Fujifilm LAS-4000 imager. Image analysis was performed using LI-COR Image Studio Lite. Each dot was assigned a value, and the average of two measurements was calculated for each array. The average density values for each cytokine in each condition were then averaged for each array to calculate the fold change. IL-6, IL-7, and IL-8 levels were quantified using IL-6, IL-7, and IL-8 ELISA kits, respectively, according to the manufacturer’s instructions (R&D Systems, Minneapolis, MN, USA).

### 4.6. Melanin Level Determination and Tyrosinase Enzymatic Activity Assay

To measure cellular tyrosinase activity, equal amounts of cell lysates (10 μg) were incubated with 10 mM L-dihydroxyphenylalanine (L-DOPA) (pH 6.8) (Sigma-Aldrich, St. Louis, MO, USA) at 37 °C for 1 h. The amount of melanin produced from L-DOPA via tyrosinase activity in the cell extracts was measured using a UV-vis spectrometer (Molecular Devices, San Jose, CA, USA) at 490 nm. To determine the cellular melanin levels, the cell pellets were dissolved in 50 μL of 1 N sodium hydroxide and the melanin levels were determined by measuring the absorbance at 490 nm. The melanin levels were normalized to the protein input of the samples.

### 4.7. RT-qPCR

Total RNA was isolated using TRIzol reagent (Invitrogen, Carlsbad, CA, USA) according to the manufacturer’s instructions. The RNA concentration was determined spectrophotometrically, and the integrity of the RNA was assessed using Bioanalyzer 2100 (Agilent Technologies, Santa Clara, CA, USA). Two micrograms of RNA was reverse-transcribed into cDNA using SuperScript III reverse transcriptase (Invitrogen), and aliquots were stored at −20 °C. TaqMan RT-PCR technology (7500Fast, Applied Biosystems, Foster city, CA, USA) was used to determine the expression levels of selected target genes with TaqMan site-specific primers and probes. The process included a denaturing step performed at 95 °C for 10 min, followed by 50 cycles of 95 °C for 15 s and 60 °C for 1 min. The following probes were used: il-6 (Hs00985639_m1), il-8 (Hs00174103_m1), tnfr1 (Hs01042313_m1), adiponectin (Hs00605917_m1), il-7 (Hs00174202_m1), tgf-β (Hs00820148_g1), ifn-γ (Hs00989291_m1), mitf (Hs01117294_m1), tyr (Hs00165976_m1), tyrp1 (Hs00167051_m1), dct (Hs01098278_m1), mart1 (Hs06637009_s1), ckit (Hs00174029_m1), sox10 (Hs00366918_m1), scf (Dm01795022_g1), and gapdh (Hs02786624_g1), which was used as a housekeeping gene.

The reactions were performed in triplicate. The mRNA expression levels were quantified using the relative *C*_T_ method and were normalized to *GAPDH* levels.

### 4.8. Western Blot Analysis

To prepare cell lysates, NHEKs and human epidermal melanocytes were washed with ice-cold PBS and lysed in RIPA buffer (50 mM Tris-HCl pH 7.4, 150 mM NaCl, 0.5% sodium deoxycholate, 0.1% SDS, and 1% NP-40) in the presence of a protease and phosphatase inhibitor cocktail (Sigma). The lysates were then centrifuged at 15,000× *g* for 15 min, and the supernatants were used for analysis. The protein concentrations were determined using a BCA kit (Sigma) using bovine serum albumin (BSA) as the standard. Equal amounts of protein (30 μg/well) from cell lysates were loaded, separated using 4–12% gradient SDS-PAGE, transferred to PVDF membranes, and incubated with appropriate antibodies for 1 h. This incubation was followed by incubation with precleared protein G beads (GE Healthcare, Madison, WI, USA) overnight at 4 °C. Next, the beads were washed five times with lysis buffer. Western blotting was performed following standard protocols. The cell lysates were boiled in SDS sample buffer and resolved using 4–12% SDS-PAGE. The proteins were then transferred to a PVDF membrane (Invitrogen) and probed using specific antibodies (1:200 dilutions for all antibodies).

### 4.9. Determination of Intracellular ROS Levels and Antioxidant Activity

Cells were seeded in 24-well plates for 24 h and then treated with UVB and IPL radiation for 18 h. Intracellular ROS levels were measured by DCFDA assay (excitation: 485 nm; emission: 535 nm) with a microplate reader (SPECTROstar Nano, BMG Labtech, Ortenberg, Germany).

Catalase activity was examined in 50 μg of protein from cell lysates with an assay solution containing 0.01 M phosphate buffer (pH 7.0) and 0.015 M hydrogen peroxide in a final volume of 1 mL. Catalase activity was evaluated according to the change in the optical density at 570 nm with a microplate reader and a catalase activity assay kit (ab83464, Abcam, Cambridge, UK). One unit of enzyme activity was defined as the amount of the enzyme required to scavenge hydrogen peroxide per minute under defined conditions. SOD activity was determined at 450 nm via inhibition of tetrazolium salt reduction; a formazan dye was produced upon reduction with a superoxide anion (HT Superoxide Dismutase Assay Kit, R&D Systems, Minneapolis, MN, USA).

Resistance to oxidative stress was calculated by estimating the cell survival rate under hydrogen peroxide treatment conditions (0.1 M, 4 h) with a Cell Counting Kit-8 (Dojindo, Japan).

### 4.10. Statistical Analysis

Data are expressed as the mean ± SD. The normality of data was analyzed using the Shapiro–Wilk test, and results between different groups were compared using one-way ANOVA (followed by Dunnett’s post hoc test) or Student’s *t*-test. For RT-qPCR, data are shown as the mean ± SD of at least three triplicate measurements. The *p*-values generated via two-tailed Student’s *t*-test were used to compare Δ*C*_T_ values between the control and treatment groups. All statistical tests were two sided, with the level of significance established at *p* < 0.05. SPSS software (ver. 21; SAS Institute, Cary, NC, USA) was used for statistical analyses.

## Figures and Tables

**Figure 1 ijms-22-03173-f001:**
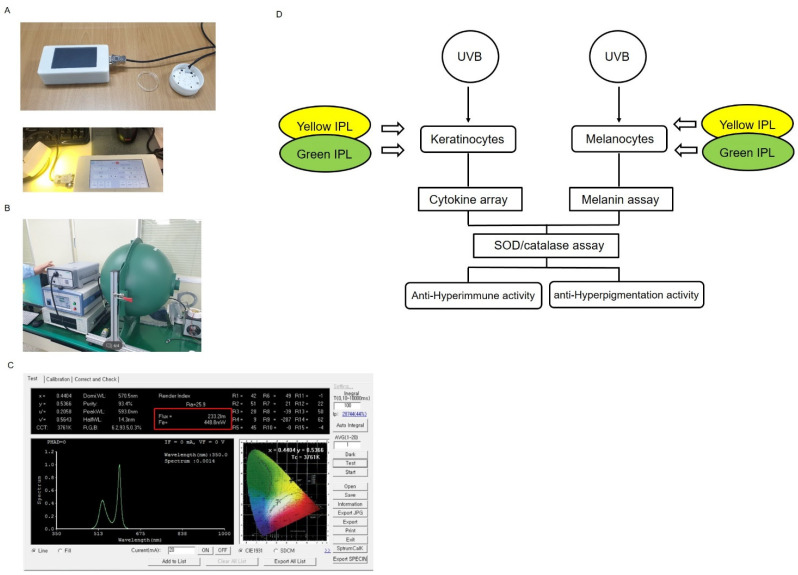
Intense pulse light (IPL) device settings and outputs. (**A**) Energy meter with a handheld display unit (left), a sensor unit (right), and a 60 mm^2^ (actual inside growth surface diameter: 52.1 mm) plate well assembled in the unit shield. (**B**) A small integrating sphere system and high-accuracy spectroradiometer were used to measure the IPL outputs. (**C**) The device setting display shows the lumens, currents, and duty cycles of yellow and green IPL. (**D**) The schematic diagram shows the process of study.

**Figure 2 ijms-22-03173-f002:**
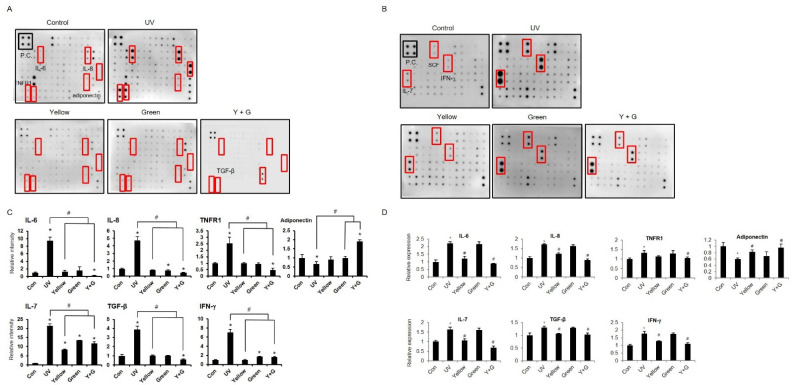
IPL treatment attenuates the ultraviolet B (UVB)-induced hyperimmune response. (**A**,**B**) IPL-treated normal human epidermal keratinocyte (NHEK) culture medium was harvested for a cytokine array. Representatively changed cytokines and positive control spots are highlighted. (**C**) The relative intensity of cytokines was represented. (**D**) The relative expression of cytokines that were altered in the array was measured by RT-qPCR. The results are shown as the means ± SDs of the values obtained in 3 independent experiments (N = 3). # *p* < 0.01 compared to the UVB-treated group; * *p* < 0.001 compared to the control group.

**Figure 3 ijms-22-03173-f003:**
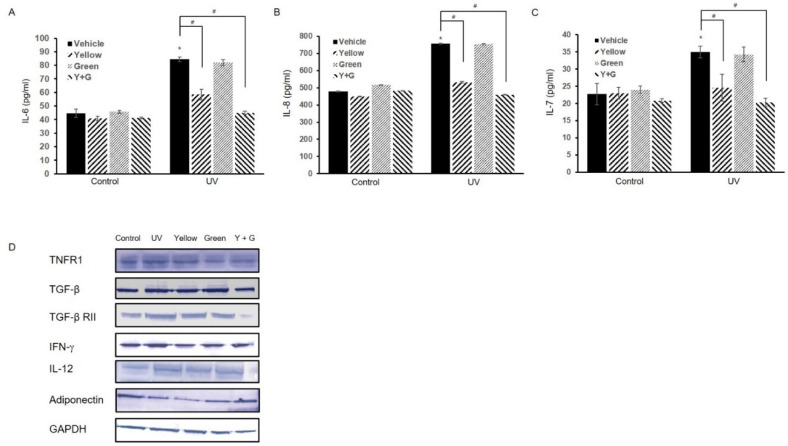
IPL application attenuates proinflammatory cytokine secretion and protein expression induced by UVB irradiation. Cell lysates were applied to analysis with specific ELISA kits to detect secreted (**A**) IL-6, (**B**) IL-8, and (**C**) IL-7. (**D**) The protein expression of tumor necrosis factor receptor 1 (TNFR1), transforming growth factor beta (TGF-β), TGF-β RII, interferon gamma (IFN-γ), interleukin (IL)-12, and adiponectin was detected by Western blotting. The full blotting data are presented in Appendix A. The data are presented as the means ± SDs of the values obtained in 3 independent experiments (N = 3). # *p* < 0.001 compared to the UVB-treated group; * *p* < 0.001 compared to the vehicle-treated (control) group.

**Figure 4 ijms-22-03173-f004:**
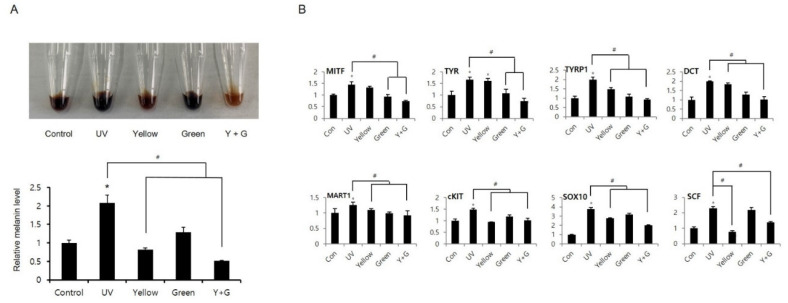
IPL treatment inhibits melanogenesis caused by UVB radiation in human epidermal melanocytes. Melanocytes were pretreated with UVB, and then IPL irradiation was performed. (**A**) The melanin accumulation levels were represented by the colors of the cell lysates, and the melanin content in the cell lysates was analyzed via spectrometry. (**B**) The expression of melanogenic genes was examined by RT-qPCR. The data are depicted as the means ± SDs of the values obtained in 3 independent analyses (N = 3). # *p* < 0.001 compared to the UVB-treated group; * *p* < 0.001 compared to the vehicle-treated (control) group.

**Figure 5 ijms-22-03173-f005:**
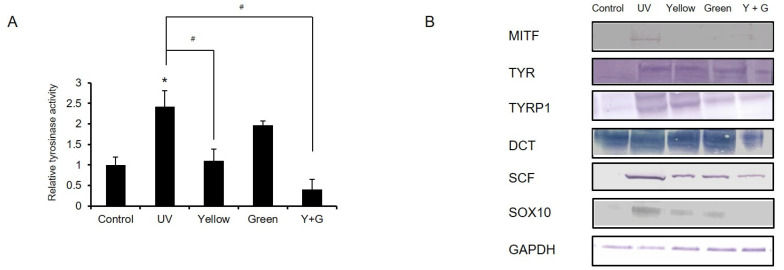
Anti-melanogenic effect of IPL on UVB-induced tyrosinase activity and overexpression of melanogenic proteins in human epidermal melanocytes. (**A**) The UVB-induced increased in tyrosinase activity was attenuated by yellow IPL treatment and combined yellow and green IPL treatment. (**B**) The expression of the melanogenic proteins microphthalmia-associated transcription factor (MITF), tyrosinase (TYR), tyrosinase-related protein 1 (TYRP1), dopachrome tautomerase (DCT), stem cell factor (SCF), and SRY-box transcription factor 10 (SOX10) was investigated by Western blot analysis. The full blotting data are presented in Appendix A. The results are presented as the means ± SDs of the values obtained in 3 independent experiments (N = 3). # *p* < 0.001 compared to the UVB-treated group; * *p* < 0.001 compared to the control group.

**Figure 6 ijms-22-03173-f006:**
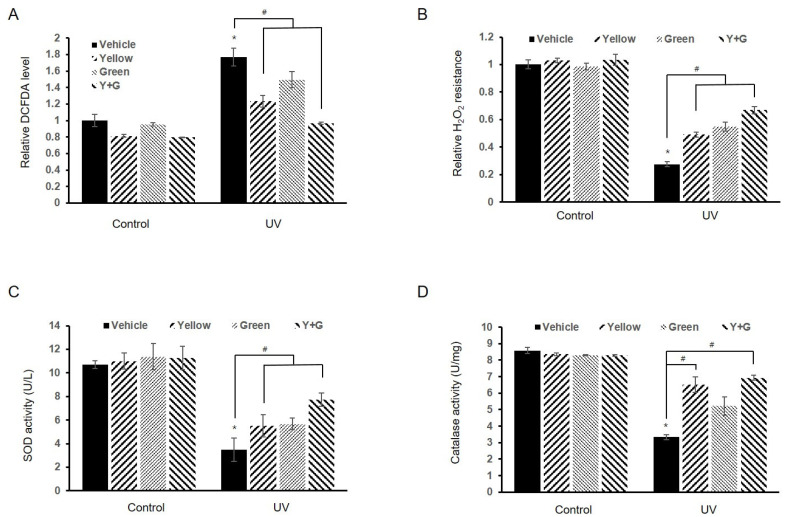
IPL reduces cellular oxidative stress levels and results in retention of antioxidative enzyme activity under UVB irradiation conditions. (**A**) Cellular reactive oxygen species (ROS) levels in the control group and the UVB-treated group were measured using a 2’,7’-dichlorofluorescein diacetate (DCFDA) assay. (**B**) Resistance to oxidative stress caused by hydrogen peroxide stimulation was investigated under IPL treatment conditions. (**C**) Superoxide dismutase (SOD) activity was examined in units per liter, and (**D**) catalase activity was evaluated in units per mg in NHEK lysates using an antioxidative enzyme assay. # *p* < 0.001 compared to the UVB-treated vehicle group; * *p* < 0.001 compared to the control vehicle group.

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
