# Peer review of "Intense Pulsed Light Attenuates UV-Induced Hyperimmune Response and Pigmentation in Human Skin Cells"

_ijms, 2021, doi:10.3390/ijms22063173_

Round 1

Reviewer 1 Report

Kim et al. described the effects of green and yellow IPL treatments on both keratinocytes and melanocytes following UVB radiation. The authors demonstrated that IPL treatments both showed reduced excessive cutaneous immune response and prevented hyperpigmentation. The manuscript was clearly written and easy to follow. Here are additional remarks, some which are critical and need to be addressed:

  1. A simple schematic of the study design should be included in fig. 1 to show experimental setup and workflow would greatly improve readability.
  2. Fig. 2A & 2B. Many of the changes in intensities of the different the spots are not always apparent compared to the control due to the different exposures between different arrays. Authors should quantify the 'relative intensities' in a bar graph. Additionally, it is not clear which spots are TNFR1 and TGF-beta.
  3. It is not very clear how authors performed cell viability measurements in fig S2
  4. Fig. 3D: Changes in protein abundance does not seem to correlate to the changes detected from ELISA and gene expression. Additionally, in fig. S3 & S4, GAPDH should be shown for each of the corresponding protein blots, instead of a separate blot. 
  5. Figure 4: While the melanogenic genes expression of the combined IPL treatment are somewhat comparable to the control, can the author explain the further reduction in pigmentation level of the combined IPL treatment?

Author Response

Reviewer #1:

Comments:

  1. We added a simple schematic of the study design as Fig. 1D.

  1. We represented the relative intensities as Fig. 2C. The results of mRNA expression levels indicated in Fig. 2D. In addition, the spots of TNFR1 and TGF-beta clearly indicated as separate squares.

  1. The cell viability measured under treatment UVB alone or IPL alone for estimate cellular toxicity of UVB or IPL. We added the methods in materials & methods section (Line 317-318) and we rewrite the title of Fig.S2 for clarity.

  1. In skin keratinocyte, it is hard to detect the expression of pro-inflammatory proteins. Actually, we could not detect IL-6, 7, and 8 despite of multiple trial. Although protein abundance does not seem to correlate clearly, the results of protein abundance with western blotting shown in Fig. 3D showed same tendency with cytokine array result. We tried multiple times and we apologize to the readers and reviewers that we could not represent the clear results. Additionally, we performed the western blot analysis with one protein in one membrane for the obvious results and do not use same membrane again. We showed GAPDH protein blot with same sample of other protein. Please understand it. Thank you for your understanding.

  1. Figure 4: We confirmed the further reduction in melanin level of the combined IPL treatment. The expression of melanogenic genes (Fig. 4B) and tyrosinase activity (Fig. 5A) also indicated that combined IPL attenuated melanogenesis. IPL has been used therapeutically in a clinical setting to reduce skin pigmentation not only in pigmented lesions but also in normal skin. Because, the settings in these cases have similar settings with yellow or green IPL in skin therapy, it is possible combined IPL treatment could attenuated melanogenesis in normal melanocytes.

Reviewer 2 Report

This is an interesting manuscript presented by Juewon Kim and colleagues.

I have a few comments:

Abstract

Line 19

The below sentence is not conclusive. If authors can be double-check:

“We found that IPL treatment reduced excessive cutaneous immune 18 reactions by suppressing UVB-induced increases in inflammatory cytokine levels. “

Introduction

 Line 29-48

First paragraph of introduction somewhat repetitive. Not all statements factual correct, for instance sentence in line 42/43: “UVB exposure elevates the generation of a variety of inflammatory cytokines associated with DNA repair,….” .

DNA repair does not cause the generation of inflammatory cytokines. A rephrasing of this sentence might be advised.

Line 54/55
It is not clear how hyperpigmentation and/or aberrant melanin production can harm the epidermal barrier (as not a malignant process). Melanocytes reside in the basal layer of the epidermis.

Results

Line 110

Methods do not mention the investigation of porcine skin; authors might like to add this to the methods.

Section 2.2.

It is not clear, how the control is being treated and at which time point control samples have been collected for analyses; if authors can please elaborate?

I am curious to learn if authors have used keratinocyte-derived supernatant to treat melanocytes to study effects of pro-inflammatory cytokines on melanin synthesis. Alternatively, have authors conducted co-culture experiments?

Section 2.4 /figure 6

Usage of term “vehicle” not clear. I apolgise if I missed this, but did authors evaluated melanocyte viability before, while and after treatments and compared outcomes to antioxidative responses observed?

It is not clear which cells have been used for the assays in figure 6A-D: please check caption for figure 6.D

Discussion

Line 234

This study was conducted in keratinocytes and melanocytes separately; If authors have conducted co-cultures, data should be shown.

Figure 2 does not present findings of “melanogenetic gene expression” (in keratinocytes), and figure 4 does not show results for inflammatory gene expression (in melanocytes).

Line 251

I wonder how cultured cells can exhibit photoreceptors (references 32-34)

Line 265

Melanin production as a consequence of UV (B) exposure is a highly conserved, perfectly orchestrated and essential step in maintaining the integrity of the human skin. Melanin is a highly potent photoprotective agent, as it interacts with free radicals in the cell.

Line 273:

Which “skin cells” are meant here? Keratinocytes or melanocytes?

Can authors discuss consequences of the hyperimmune reaction shown in figure 2? Increased levels of pro-inflammatory cytokines expressed by keratinocytes might affect a wealth of other cells in the skin. Which cells will be affected, and which consequences do authors expect?

Methods:

It is not clear how melanocytes have been treated (UVB/ IPL)

Antibody concentrations/dilutions should be provided and details for TaqMan probes listed.

Author Response

Reviewer #2:

Comments:

  1. Abstract Line 19: We corrected the sentence more conclusively as,

“We found that IPL treatment reduced excessive cutaneous immune reactions by suppressing UVB-induced proinflammatory cytokine expressions.”

  1. Introduction:

We rephrasing and rewrite the sentences as underlined paragraphs.

  1. Line 54/55: We corrected the sentence as,

Aberrant melanin production due to photoaging can be harmful to skin health.

Results

  1. Line 110: We described the experiment in Figure S1 with methods.

  1. section 2.2: control is not treated; UV group: 18 hours of UVB irradiation (20 mJ/cm2). We set the UVB irradiation conditions based on data on cell viability and the levels of a representative UV irradiation-induced inflammatory cytokine, interleukin (IL)-6 (Figure S2A-B). IPL group: yellow, green, or yellow + green with settings of 150 lumen, 100 ms pulse duration, and 30 min interval time for 18 hours. We set the IPL conditions according to cell viability and cytotoxicity experiments (Figure S2C-D). After these treatment, all experimental samples have been collected for analysis, at the same time point.

  1. We performed the study with keratinocytes or melanocytes, separately. We investigated the effects of IPL on the UVB-irradiated keratinocytes or UVB-treated melanocytes.
  2. section 2.4/figure 6:

7-1. “vehicle” means “not treated”. Because experimental groups named as “control” and “UV”, we named the not treated groups as “vehicle” group. Actually, “vehicle” group not treated with IPL but only in media.

7-2. The experiments in figure 6 were performed with normal human epidermal keratinocytes as indicated in figure 6 captions. The cell viability with UV or IPL treatment assessed in Figure S2.

Discussion

  1. Line 234: We performed this study in keratinocytes or melanocytes, separately. We did not conduct co-cultures experiments. We expect further study.

  1. Figure 2 and Figure 4: It is interesting point to evaluate the expression of melanogenic genes in keratinocytes or inflammatory genes in melanocytes. Unfortunately, melanogenic genes showed quiet low expression in keratinocytes, and inflammatory genes abundance is merely confirmed in melanocytes. Thank you for your advice.

  1. Line 251: Many studies with light therapy were carried out human or animal model to investigate the light effects. But also, in humans, opsins are present in various skin cell types, including keratinocytes, melanocytes, dermal fibroblasts, and hair follicle cells (Wicks et al., Curr Biol. 2011; 21(22): 1906-1911; Haltaufderhyde et al., Photochem Photobiol. 2015; 91(1): 117-123). They have been shown to mediate wound healing, melanogenesis, hair growth, and skin photoaging. We added the references as ref. 35-36. Study about the role of photoreceptors in cultured cell is interesting theme for future research.

  1. Line 265: We totally agree with your opinion. Melanin production as a consequence of UVB exposure is a highly conserved process and it interacts with free radical in the cell. We thought IPL treatment regulates redox state and result in melanogenesis modulation.

  1. Line 273: We indicated skin cells as

“UVB-induced hyperimmune in keratinocytes and melanogenic responses in melanocytes are related to cellular redox levels and these changes affects each other”. Keratinocytes promotes melanogenesis, melanocyte proliferation, and dendrite formation of melanocytes through the production and release of factors which related in immune reaction, such as TNF-α, interleukins, and also stem cell factor (SCF) and these genes are stimulated by UV rays (Wang et al., J. Allergy Clin Immunol. 2017; 139, 12059-1216). We also added the paragraph.

  1. Methods: We added how melanocytes have been treated in 4.4 section. Actually, treatment condition applied same as keratinocytes and melanocytes. We indicated the dilutions of antibody for western blot analysis (4.8.) and provided TaqMan probes information (4.7.).

We UNDERLINED the location of revisions in the manuscript. And we enlarged and corrected all figures, according to journal format.

Sincerely yours,

Juewon Kim, Ph. D.

Round 2

Reviewer 1 Report

Thank you for the revised manuscript. It is extremely unfortunate that the authors did not re-blot each of the proteins against GAPDH protein to show appropriate loading control. I anticipate that this will be one of the major concerns most readers will pick up on. Here are few minor comments to improve the ecstatic on the updated version:

  1. Better 'framing' for Fig. 3D for IL-3 to show full band for Y+G
  2. Better 'angle' for Fig. 5B for TYRP1 to show complete bands in all lanes.